# Vertebral Body Height Changes in Acute Symptomatic Osteoporotic Vertebral Compression Fractures Treated with Vertebral Cement Augmentation—Which Factors Affect Vertebral Body Height during Follow-up? A Multiple Linear Regression Study

**DOI:** 10.3390/geriatrics7060142

**Published:** 2022-12-14

**Authors:** Jesús Payo-Ollero, Rafael Llombart-Blanco, Carlos Villas, Matías Alfonso

**Affiliations:** Orthopaedic Surgery and Traumatology Department, Clínica Universidad de Navarra, Av. Pio XII 36, 31008 Pamplona, Spain

**Keywords:** aging spine, spine surgery, osteoporotic fracture, back pain, vertebroplasty

## Abstract

Changes in vertebral body height depend on various factors which were analyzed in isolation and not as a whole. The aim of this study is to analyze what factors might influence the restoration of the vertebral body height after vertebral augmentation. We analyzed 48 patients (108 vertebrae) with osteoporotic vertebral fractures who underwent vertebral augmentation when a conservative treatment proved to be unsatisfactory. The analyses were carried out at the time of the fracture, during surgery (pre-cementation and post-cementation), at the first medical check-up (6 weeks post-surgery) and at the last medical check-up. The average vertebral height was measured, and the differences from the preoperative values were calculated at each timepoint. A Pearson correlation coefficient and a linear multivariable regression were carried out at different timepoints. The time since the vertebral fracture was 60.4 ± 41.7 days. The patients’ average age was 73.8 ± 7 years. The total follow-up period was 1.43 ± 1 year. After vertebral cementation, there was an increase in the vertebral body height of +0.3 cm (13.6%). During the post-operative follow-up, there was a progressive collapse of the vertebral body, and the pre-surgical height was reached. The factors that most influenced the vertebral height restoration were: a grade III collapse, an intervertebral-vacuum-cleft (IVVC) and the use of a flexible trocar before cement augmentation. The factor that negatively influenced the vertebral body height restoration was the location of the thoracolumbar spine.

## 1. Introduction

Different meta-analyses have compared percutaneous vertebroplasty (PVP) with the conservative treatment for osteoporotic vertebral compression fractures (OVCFs), reaching the conclusion that PVP achieves greater pain relief, greater functional recovery and better quality of life during the first post-operative year [1,2,3].

Although vertebral height restoration is not an objective of PVP, numerous studies have shown that there is an increase in the vertebral body height after PVP of 1.2–2.3 mm [4,5,6,7,8,9]. However, the increase in the vertebral body height is frequently calculated comparing the pre- and postsurgical measurements, which means that the changes that occur because of the patient’s position (from standing to prone positions) are not taken into account, and this could bias the vertebral augmentation effect [4,7,9,10,11].

An increase in the vertebral body height depends on different factors. Dynamic mobility, which is conditioned by the patient’s position, is the change in the vertebral body height when the patient goes from standing to supine or prone positions, and it constitutes one of the most important factors [8,12,13,14,15,16]. The type of vertebral fracture (wedge, biconcave or burst) determines which area of the vertebral body (anterior, middle or posterior) experiences greater mobility [7,9,15]. Vertebral fractures of the thoracolumbar region have greater mobility than those in the thoracic region or lumbar region do [14,15]. Percutaneous kyphoplasty (PKP) achieves a greater restoration of vertebral height than PVP does [5,6]. The presence of an intervertebral vacuum cleft (IVVC) is another important factor [8,9], and these clefts are usually located in the thoracolumbar region. All of these factors are generally analyzed in isolation and not as a whole, so surgeons do not really know when these factors may exert an influence during the patient follow-up.

For years, the relationship between osteoporotic vertebral fractures and sagittal balance changes has been proven. Isolated vertebral compression fractures change the local parameters. However, if there are multiple fractures, they might cause a sagittal spinal imbalance [17,18]. Dorsal hyperkyphosis and the loss of vertebral height create an anterior displacement of the gravity center. These alterations might cause respiratory problems, gastrointestinal disturbances, and ultimately, spinal canal compression. In the anterior body vertebral segment, a flexion moment that is significant is created, and the posterior muscles and ligaments must compensate for this flexion excess, and the anterior body vertebral segment that is weakened must resist more compressive forces. This leads to impaired balance, increased muscle fatigue, reduced gait speed, an increased risk of falls, and therefore, an increased risk of additional fractures [19]. A recent study by Su et al. [20] studied the effect of PVP on sagittal alignment in 42 patients. They found that two weeks after the treatment pain, their function and sagittal balance improved: sagittal vertical axis (from 49 to 37.75), T1 pelvic angle (from 22.98 to 20.69) and thoracic kyphosis (from 25.83 to 14.91). It is also important to be aware that the previous presence of a sagittal imbalance is a risk factor for vertebral re-collapse after PVP [21].

The aim of this study is to specify the factors influencing the restoration of the vertebral body height after PVP from the vertebral body fracture until the patient is definitively discharged.

## 2. Materials and Methods

We conducted this observational, prospective and non-randomized study in compliance with the principles of the Declaration of Helsinki. The study’s protocol was reviewed and approved by the Institutional Review Board of University of Navarra (IRB No. 2018.044). Written informed consents were obtained from all of the patients.

### 2.1. Patient Selection

Our cohort consisted of patients who had been treated for acute painful OVCFs and who met the inclusion criteria over the period from January 2015 to May 2020. The recruitment criteria were: patients older than 65 years old, having acute painful OVCFs, the failure of conservative treatment (non-steroidal anti-inflammatory drugs, rigid orthosis and lumbar exercises), the progressive collapse of the vertebral body, reproducible pain at the level of the fractured vertebra, with either a current MRI showing bone edema on the STIR sequence or cement augmentation without instrumentation. The patients were excluded if they did not meet inclusion criteria, which were: if they had a vertebral fracture secondary to the oncological disease or if the cement augmentation was associated with instrumentation. Of 50 patients (117 vertebrae) treated with cement augmentation, 2 of them (9 vertebrae) were excluded because they were lost to the follow-up. Finally, 48 patients were included. Thirty-four patients (78 vertebrae) were treated with percutaneous vertebroplasty (DePuy Synthes, Oberdorf, Switzerland), ten patients (16 vertebrae) were treated with a flexible trocar StabiliT^®^ MX Vertebral Augmentation System (DFine, San José, CA, USA) and four patients were treated with both of the techniques (10 vertebrae with percutaneous vertebroplasty and 4 vertebrae with StabiliT^®^ system). These four patients were treated at two different timepoints. All of the patients underwent conventional AP/Lateral X-Rays as well as MRI, pre-surgery. The location of the treated vertebrae was as follows: T5 (*n* = 2), T6 (*n* = 9), T7 (*n* = 9), T8 (*n* = 10), T9 (*n* = 6), T10 (*n* = 6), T11 (*n* = 5), T12 (*n* = 10), L1 (*n* = 10), L2 (*n* = 12), L3 (*n* = 10), L4 (*n* = 10) and L5 (*n* = 9). The demographic parameters of our cohort are shown in Table 1.

### 2.2. Description of the Surgical Technique

The surgical treatment was performed under general anesthesia in a prone position. We did not perform any maneuver to restore the vertebral height before or during the procedure. Two imaging systems (SIEMENS Arcadis Orbic and SIEMENS Arcadis Varic) perpendicular to each other and centered on each vertebra were used. All of the vertebroplasties were performed through a bilateral transpedicular approach. Under fluoroscopic guidance, cement was injected. The cement injection procedure continued until the vertebral body was filled toward the posterior 25% of the vertebral body or until leakage occurred. After the cement injection, the patient remained prone on the table for approximately 20 min.

The differences between using the StabiliT^®^ system and PVP were: before injecting them with the cement, a path was created with a flexible trocar, the administration of the cement was carried out with a remote control and the cement viscosity was controlled by radiofrequency. The main benefits of the StabiliT^®^ system are: that we can guide the injected cement through the path created by the flexible trocar, reducing the volume of injected cement and controlling the viscosity of the cement due to radiofrequency.

### 2.3. Collected Data

All of the patients were studied pre-surgery, pre-cementation, post-cementation, at the first medical check-up (6 weeks post-surgery) and at the last medical check-up (minimum 6 months). A lateral X-ray was used to take the measurements. All of the images were taken at a distance of one meter from the patients, ensuring that the upper and lower plate were properly aligned. In each vertebral body, the anterior, middle (most collapsed zone and least collapsed zone) and posterior edge heights were measured (Figure 1). The vertebral height was the average of each part of the vertebral body. In each period, the difference was calculated from the preoperative values. The posterior edge of one adjacent nonfractured vertebral body was also measured in each study period to validate the measurements. The first and second authors made the measurements using a digital PACS caliper, and they were blinded to clinical context in each case.

The other assessed parameters were the demographic data (age, gender, body mass index (BMI) and T-Score), the vertebral fracture evolution time (less or more than 6 weeks), the type and severity of vertebral fracture according to the Genant classification [22], an IVVC presence, the vertebral augmentation technique and the volume of cement injected. Vertebral fracture location was classified as follows: thoracic (T1–T10), thoracolumbar (T11–L2) or lumbar (L3–L5).

### 2.4. Statistical Analysis of Data

The sample size was determined by two means, taking into account the data published in the article by Röllinghoff et al. [23] This study was chosen because the methodology is similar to our study. The mean and standard deviation of the pre-surgical and post-surgical vertebral body heights were chosen for those vertebrae that were treated by the vertebroplasty and StabiliT^®^ system (vertebroplasty, 14.1 ± 5.1 mm and 17.9 ± 4.2 mm; StabiliT^®^ system, 14.7 ± 5.6 mm and 19.5 ± 4.5 mm). The beta was 0.2 (80% power), and the alpha (two-tailed) 0.05. We determined that a minimum of 24 vertebrae was needed if we took into account the results of PVP, or we needed 18 vertebrae if we took into account the results of StabiliT^®^ system.

Descriptive statistics about the sample were obtained. The Shapiro–Wilk test confirmed normal distribution of the variables. Initially, a Pearson correlation coefficient was carried out between the demographic quantitative variables (age, BMI and *t*-Score) and the difference in overall height of the vertebral body at the different study timepoints regarding the pre-surgical status. We performed an intraclass correlation coefficient for evaluating the concordance between the first and second authors. The gender variable was analyzed using Student’s *t* test at the different timepoints.

Subsequently, we performed a multiple linear regression to determine if the vertebral fracture time, vertebral segment treated, type of fracture treated, severity of the fracture, IVVC presence or vertebral augmentation technique performed could influence the restoration or loss of vertebral height at each timepoint. Finally, to verify the validity of our model, we checked that the residuals followed a normal distribution.

A 0.05 level of probability was accepted as the criterion for statistical significance for all of the statistical tests. All of the statistical tests were carried out using Stata software 12.0 version for Macintosh (Data Analysis and Statistical Software, TX, USA).

## 3. Results

### 3.1. Patients

The average time from diagnosis to surgery was 60.4 ± 41.7 days. Specifically, 49 vertebrae (45.4%) were treated before the sixth week and 59 vertebrae (54.6%) were treated after the sixth week. The average age of the patients was 73.8 ± 7 years. The BMI was 26.5 ± 4 kg/m^2^, and the mean T-score was −1.9 ± 1.1 (Table 1). Thoracic injury predominated, with it being found in 42 vertebrae (38.9%). According to the Genant classification [22], 75 vertebral injuries were of the biconcave type (69.4%) and 33 vertebral injuries were of the wedge type (30.6%). A grade III collapse was the most frequent one (Table 1). The average volume of injected cement was 3.5 ± 1.18 mL.

### 3.2. Radiological Measurement

The pre-surgical radiography was performed 5.8 ± 3.6 days before the procedure. After surgery, the X-rays were performed at 46.5 ± 18.1 days (first medical check-up) and 17.1 ± 12 months post-surgery (last medical check-up). The intraclass correlation coefficient was 0.79. It represented an excellent reliability between the authors.

There were no differences among each timepoint in the height of the posterior edge of one adjacent nonfractured vertebral body compared to the pre-surgical moment (pre-surgery: 3.12 ± 0.04 cm; pre-cementation: 3.11 ± 0.04 cm, difference −0.013 cm, *p* = 0.277; post-cementation: 3.1 ± 0.04 cm, difference −0.019 cm, *p* = 0.264; first medical check-up: 3.1 ± 0.04 cm, difference −0.023, *p* = 0.079; last medical check-up: 3.12 ± 0.04 cm, difference −0.001 cm, *p* = 0.911).

The pre-surgical vertebral body height was 2.2 ± 0.6, and it increased to 2.3 ± 0.5 (difference with pre-surgical status of +0.1 cm, +4.5%, 95%CI 0.034 to 0.125) with the patient’s position change on the surgical table (from standing to a prone position). Post-cementation, the vertebral body height increased to 2.5 ± 0.5 (difference with pre-surgical status of +0.3 cm, +13.6%, 95% CI 0.156 to 0.282). At the first medical check-up, the vertebral body height decreased to 2.3 ± 0.5 (difference with pre-surgical status of +0.1 cm, +4.5%, 95% CI 0.045 to 0.147). Finally, at the last medical check-up, the vertebral body height had decreased to the pre-surgical levels (2.2 ± 0.5 versus 2.2 ± 0.6, difference of +0.009 cm, +0%, 95% CI −0.044 to 0.063) (Figure 2).

Taking into account the cementation technique (PVP versus StabiliT^®^ system), we did not observe differences in the vertebral height pre-surgery (2.24 cm vs. 2.29 cm, difference 0.05 cm, 95% CI −0.331 to 0.229). We did not detected differences in the vertebral height with regard to the patient’s position, post-cementation or first medical check-up, and these differences were not statistically significant if we compared those at each point of the follow-up (*p* ≥ 0.05) (Figure 3, Table 2). However, in the last medical check-up, we observed a greater vertebral height when we were using the StabiliT^®^ system than we did with PVP (StabiliT^®^ system 2.47 cm, versus PVP 2.21 cm, difference 0.26 cm, 95% CI −0.496 to −0.008). On the other hand, if we compared each point of the follow-up with the pre-surgery data, the vertebrae treated using the StabiliT^®^ system partially maintained the vertebral height that was restored in the first (StabiliT^®^ system 0.25 cm versus PVP 0.06 cm vertebrae height restored, difference 0.18 cm, 95% CI −0.307 to −0.053) or the last medical check-ups (StabiliT^®^ system 0.18 cm versus PVP −0.03 cm vertebrae height restored, difference 0.21 cm, 95% CI −0.336 to −0.068).

### 3.3. Multiple Linear Regression

A Pearson correlation coefficient was performed to determine the relationship between the quantitative demographic variables (age, BMI and T-Score) and the difference in vertebral body height at each timepoint with respect to the pre-surgical moment. No variable was statistically correlated (*p* ≥ 0.05), and they presented a weak association, so they were excluded from the multiple linear regression. The gender variable also had no statistical differences when the height difference of the vertebral body was compared at the different timepoints (*p* ≥ 0.05), and so this was also excluded from the multivariate analysis.

We performed four multiple linear regression models, one for each timepoint (Table 3). As a dependent variable, the global difference in the vertebral body height at the time of study analyzed with respect to the pre-surgical moment was used. The vertebral segment treated, type and severity of vertebral fracture, vertebral fracture time, vertebral augmentation technique and volume of cement injected were used as independent variables.

All of the differences were statistically significant (*p* ≤ 0.001). The coefficient of determination was 0.1516 at the pre-cementation point, it was 0.2616 at the post-cementation point, it was 0.2585 at the first medical check-up and it was 0.2263 at the last medical check-up point. The different linear regression models show that the main factors influencing vertebral body height restoration were: a grade III collapse (at the pre-cementation time, post-cementation time and the first post-surgical medical check-up time), IVVC (only at the post-cementation time), and the StabiliT^®^ system (at the first and last post-surgical medical check-up). In contrast, a fracture located in the thoracolumbar region negatively influenced the vertebral body height (at the first post-surgical medical check-up) (Table 3 and Table 4).

## 4. Discussion

This study focuses on factors that influence the restoration of the vertebral body height during a vertebral fracture treatment, quantifying this from diagnosis to discharge. We observed that some factors favored the restoration of the vertebral body height, while others affected it negatively. Furthermore, these factors acted concretely at different times during the patient follow-up.

Previously published studies that analyze change in vertebral body height after vertebral augmentation have various limitations [4,6,7,9,24,25]. The irregularity of the vertebral fracture makes it difficult to select a reference point to carry out the measurements. Additionally, there is no consensus on how to quantify the changes in the vertebral body height, and the studies take into account different factors (dynamic mobility, IVVC and vertebral augmentation technique) that are analyzed in isolation and not as a whole [4,8,9,12,13,14,16,19,26]. This means that the surgeons do not know at what point in the patient follow-up that these factors influence vertebral restoration.

The overall changes in the vertebral body have previously been studied at two points in time: pre- and post-surgery [4,5,6,12,23]. To provide more detailed and accurate information, we analyzed the vertebral body at different timepoints: pre-surgery, intra-operatively (pre-cementation and post-cementation), at the first medical check-up (6 weeks) and at the last medical check-up (17.1 ± 12 months). To our knowledge, our study is the first one to analyze the changes intraoperatively, establishing the changes due to the patient’s position in order to differentiate the real effect of vertebral augmentation [4,6,7,9,10,11].

Chen et al. [12] and Yokoyama et al. [16] concluded that vertebral height restoration depends more on dynamic mobility than vertebral augmentation does. McKiernan et al. [8] observed that approximately 35% of the vertebrae were mobile. In our study, we found that after vertebral augmentation, there was an increase in the vertebral height of +0.3 cm (13.6%). Specifically, the patient’s position change (from standing to a prone position) led to an increase in the vertebral height of +0.1 cm (+4.5%), and vertebral cementation caused a further increase in the vertebral body height of +0.2 cm (+8.7%). These findings support the conclusion of Chen et al. [12] and Yokoyama et al. [16] since mobility of the vertebra contributed to a third of the height restoration, and the cementation was the cause of the rest of it.

If we assess, as a whole, the factors that influence vertebral height restoration, we can see that these factors act at different times. We observed that having a severe collapse (grade III) had the greatest influence when the patient was in the prone position (Table 3 and Table 4). This observation differed from the conclusions of McKiernan et al. [8] or Teng et al. [9] who suggested that the change in height was favored by the presence of an IVVC. However, our results show that IVVC had the greatest influence when the vertebral augmentation was performed (*p* = 0.018), and it was not as a result of the patient’s position change (*p* = 0.250). Determining to what extent and when a factor most influences vertebral height restoration is only possible if intraoperative radiographic measurements are performed.

The fractures occurring in the thoracolumbar zone have greater mobility than thoracic or lumbar fractures do [8,14,15]. However, previously published articles do not specify whether this factor favors or disfavors vertebral height restoration. Our results show that a vertebral fracture localized in the thoracolumbar area negatively influenced vertebral height restoration (Table 3 and Table 4). This result can be justified because the thoracolumbar segment supports greater biomechanical stress than the thoracic or lumbar segments do.

Röllinghoff et al. [23] concluded that the StabiliT^®^ system may obtain the stable restoration of the vertebral body height. With our results, we agree with this conclusion. We obtained a difference of 0.18 cm or 0.21 cm in the vertebral height when we compared the StabiliT^®^ system with PVP at the first medical check-up or with the last medical check-up, respectively. This result can be explained because with StabiliT^®^ system, we were able to fill the cavities created by the flexible trocar, preventing or hindering the vertebral body collapse.

Tang et al. [26] and Takahashi et al. [27] suggested that the vertebral fracture time evolution influences the clinical and radiological results. The vertebral height and vertebral kyphotic angle after kyphoplasty were better in the patients who were treated in the first two months from the onset of the symptoms. Based on our results, we cannot affirm that the time that elapsed had since the vertebral fracture influenced the vertebral body height.

The outcomes in our cohort should be interpreted with caution due to the potential limitations of the study. We did not compare PVP with PKP. The literature shows that PKP requires more surgical time (which is harmful in elderly patients), and it is more expensive, which is why several studies recommend using PVP over PKP [26,28,29]. The measurements were made with radiographs. We used this technique because it is easy to use, it can be performed intra-operatively and it allows us to study the vertebral body in different positions (standing and prone positions). The gold-standard technique for making measurements is CT scanning. However, few hospitals can perform intraoperative CT scans, and in addition, it would mean exposing the patient to higher doses of ionizing radiation. We used two techniques of vertebral cementation (PVP and StabiliT^®^ system). We did not study the relationship between the vertebral height restoration and the clinical results. There may be other factors which influence the restoration of the vertebral height that we have not collected. To understand the factors involved in changes in the vertebral body height better, it would be interesting to make a comparison between a conservatively treated group and a surgical group. However, our present aim was to focus exclusively on the vertebrae that were treated with vertebral augmentation.

## 5. Conclusions

Our study found that the changes in the patient’s position (from standing to a prone position) and vertebral augmentation produced an increase in the vertebral body height. After surgery, there is a progressive collapse of the vertebral body that returns to the pre-surgical values. The main factors that favor vertebral height restoration are: a grade III collapse, IVVC and the vertebral augmentation technique. In contrast, the fracture having a thoracolumbar location influenced the vertebral height restoration negatively.

## Figures and Tables

**Figure 1 geriatrics-07-00142-f001:**
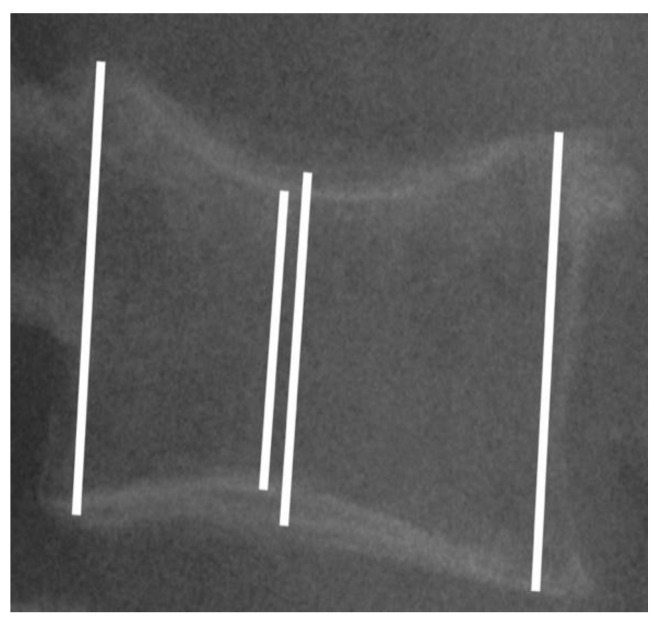
Lateral X-ray showing the measurements performed in the anterior, middle (most collapsed zone and least collapsed zone) and posterior edges. Vertebral height was the average of each part of the vertebral body.

**Figure 2 geriatrics-07-00142-f002:**
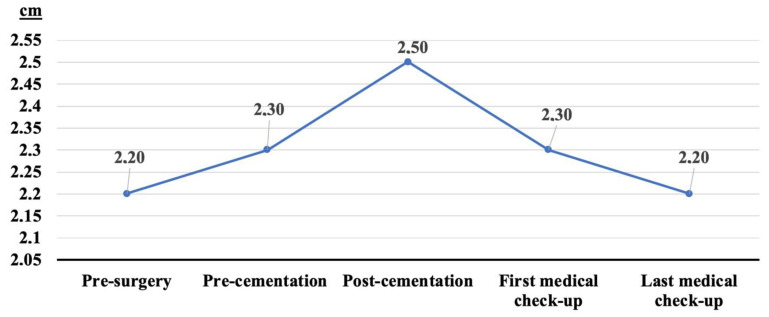
Changes in vertebral body height during follow-up. First medical check-up at 46.5 ± 18.1 days from surgery. Last medical check-up at 17.1 ± 12 months from surgery.

**Figure 3 geriatrics-07-00142-f003:**
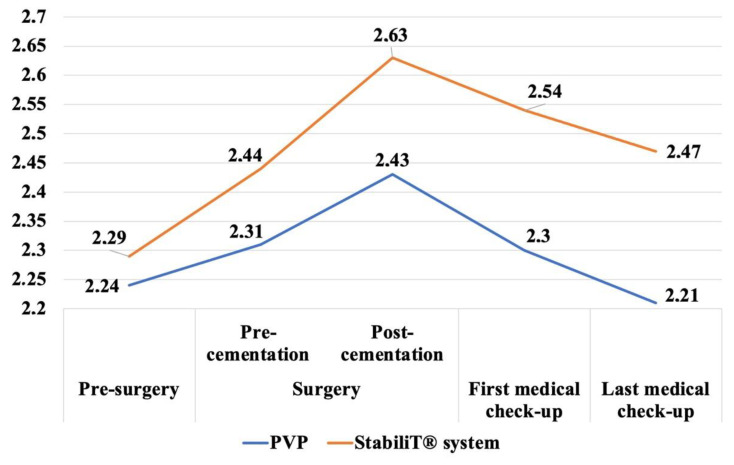
Changes in vertebral body height during follow-up taking into account vertebral technique cementation.

**Table 1 geriatrics-07-00142-t001:** Demographic characteristics.

Gender	
Male	14 (29.2%)
Female	34 (70.8%)
Age, SD	73.8 ± 7
BMI (kg/m^2^), SD	26.5 ± 4
T-Score, SD	−1.9 ± 1.1
Vertebral fracture age	
<6 weeks	49 (45.5%)
>6 weeks	59 (54.6%)
Vertebral segment treated	
Thoracic (T1–T10)	42 (38.9%)
Thoracolumbar (T11–L2)	37 (34.3%)
Lumbar (L3–L5)	29 (26.8%)
Type of fracture *	
Wedge	33 (30.6%)
Biconcave	75 (69.4%)
Severity of fracture *	
Grade I	39 (36.1%)
Grade II	29 (26.9%)
Grade III	40 (37%)
IVVC	
No	85 (78.7%)
Yes	23 (21.3%)
Volume of cement injected	3.5 ± 1.18

BMI: Body Mass Index; SD: Standard Deviation; IVVC: Intervertebral vacuum cleft; * According Genant classification [22].

**Table 2 geriatrics-07-00142-t002:** Vertebral height changed (cm) in each period between percutaneous vertebroplasty versus StabiliT^®^ system.

	Pre-Surgery	Surgery		First Medical Check-Up	Last Medical Check-Up
		Pre-Cementation	Post-Cementation		
Percutaneous vertebroplasty	2.24	2.31	2.43	2.3	2.21
StabiliT^®^ system	2.29	2.44	2.63	2.54	2.47
Difference	0.05	0.13	0.2	0.24	0.26
*p*-Value	0.723	0.322	0.135	0.061	0.042

**Table 3 geriatrics-07-00142-t003:** Multiple linear regression for the estimation of the overall vertebral body height at each timepoint with respect to the pre-surgical timepoint.

	Coefficient	SD	t	*p* Value	95% CI
Pre-cementation point					
Vertebral segment treated					
Thoracic (T1–T10)	*Ref.*				
Thoracolumbar (T11–L2)	−0.03	0.05	−0.64	0.524	−0.14 to 0.72
Lumbar (L3–L5)	0.03	0.06	0.48	0.633	−0.08 to 0.14
Type of fracture					
Wedge	*Ref.*				
Biconcave	−0.02	0.051	−0.50	0.617	−0.13 to 0.08
Severity of fracture					
Grade I	*Ref.*				
Grade II	−0.023	0.057	−0.41	0.686	−0.13 to 0.09
Grade III	0.155	0.05	2.83	0.006	0.045 to 0.26
IVVC					
No	*Ref.*				
Yes	0.06	0.05	1.16	0.250	−0.05 to 0.18
Vertebral fracture age					
<6 weeks	*Ref.*				
>6 weeks	−0.04	0.05	−0.78	0.438	−0.13 to 0.06
Constant	0.05	0.07	0.73	0.465	−0.09 to 0.20
Post-cementation point					
Vertebral segment treated					
Thoracic (T1–T10)	*Ref.*				
Thoracolumbar (T11–L2)	−0.13	0.07	−1.82	0.072	−0.27 to 0.01
Lumbar (L3–L5)	0.006	0.07	0.08	0.939	−0.14 to 0.16
Type of fracture					
Wedge	*Ref.*				
Biconcave	−0.03	0.07	−0.47	0.641	−0.17 to 0.11
Severity of fracture					
Grade I	*Ref.*				
Grade II	0.007	0.07	0.1	0.921	−0.14 to 0.16
Grade III	0.18	0.07	2.4	0.018	0.03 to 0.32
IVVC					
No	*Ref.*				
Yes	0.19	0.07	2.4	0.018	0.03 to 0.34
Vertebral fracture age					
<6 weeks	*Ref.*				
>6 weeks	0.03	0.06	0.46	0.647	−0.095 to 0.15
Vertebral augmentation technique					
Vertebroplasty	*Ref.*				
StabiliT system	0.13	0.08	1.66	0.1	−0.02 to 0.29
Cement injected	0.05	0.03	1.69	0.095	−0.01 to 0.1
Constant	−0.03	0.12	−0.22	0.826	−0.26 to 0.21
First medical check-up point					
Vertebral segment treated					
Thoracic (T1–T10)	*Ref.*				
Thoracolumbar (T11–L2)	−0.14	0.06	−2.44	0.017	−0.26 to -0.03
Lumbar (L3–L5)	−0.05	0.06	−0.74	0.464	−0.17 to 0.08
Type of fracture					
Wedge	*Ref.*				
Biconcave	−0.05	0.06	−0.93	0.354	−0.16 to 0.06
Severity of fracture					
Grade I	*Ref.*				
Grade II	−0.05	0.06	−0.88	0.379	−0.17 to 0.07
Grade III	0.15	0.06	2.57	0.012	0.03 to 0.27
IVVC					
No	*Ref.*				
Yes	0.03	0.06	0.53	0.595	−0.09 to 0.16
Vertebral fracture age					
<6 weeks	*Ref.*				
>6 weeks	−0.04	0.05	−0.68	0.496	−0.13 to 0.06
Vertebral augmentation technique					
Vertebroplasty	*Ref.*				
StabiliT system	0.22	0.06	3.46	0.001	0.09 to 0.35
Cement injected	0.0007	0.02	0.03	0.972	−0.04 to 0.04
Constant	0.12	0.09	1.23	0.221	−0.07 to 0.31
First medical check-up point					
Vertebral segment treated					
Thoracic (T1–T10)	*Ref.*				
Thoracolumbar (T11–L2)	−0.08	0.06	−1.36	0.176	−0.21 to 0.04
Lumbar (L3–L5)	−0.06	0.07	−0.92	0.358	−0.19 to 0.07
Type of fracture					
Wedge	*Ref.*				
Biconcave	−0.06	0.06	−0.97	0.334	−0.18 to 0.07
Severity of fracture					
Grade I	*Ref.*				
Grade II	−0.08	0.07	−1.24	0.219	−0.21 to 0.05
Grade III	0.11	0.06	1.65	0.102	−0.02 to 0.23
IVVC					
No	*Ref.*				
Yes	0.03	0.07	0.42	0.675	−0.11 to 0.16
Vertebral fracture age					
<6 weeks	*Ref.*				
>6 weeks	0.06	0.05	1.08	0.283	−0.05 to 0.17
Vertebral augmentation technique					
Vertebroplasty	*Ref.*				
StabiliT system	0.21	0.07	3.04	0.003	0.07 to 0.36
Cement injected	−0.01	0.02	−0.44	0.66	−0.06 to 0.04
Constant	0.03	0.11	0.35	0.726	−0.17 to 0.24

SD: Standard deviation; IVVC: Intervertebral vacuum cleft; 95% IC: 95% confidence interval; Grade I: <25%; Grade II: 26–40%. Grade III: >41%.

**Table 4 geriatrics-07-00142-t004:** Factors related to the restoration of the vertebral body height at different timepoints.

	Increase Height	Reduce Height	Questionable	Not Associated
Pre-cementation	Collapse grade III			Demographic factor (age, gender, BMI and T-Score) Vertebral segment treated Type of fracture IVVC Vertebral fracture age
Post-cementation	Collapse grade III IVVC		Thoracolumbar zone Cement injected	Demographic factor (age, gender, BMI and T-Score) Type of fracture IVVC Vertebral fracture age Vertebral augmentation technique
First medical check-up	Collapse grade III StabiliT^®^ system	Thoracolumbar zone		Demographic factor (age, gender, BMI and T-Score) Type of fracture IVVC Vertebral fracture age Cement injected
Last medical check-up	StabiliT^®^ system			Demographic factor (age, gender, BMI and T-Score) Vertebral segment treated Type of fracture Severity of fracture IVVC Vertebral fracture age Cement injected

## Data Availability

The study did not report any data.

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
