# Peer review of "Vertebral Body Height Changes in Acute Symptomatic Osteoporotic Vertebral Compression Fractures Treated with Vertebral Cement Augmentation—Which Factors Affect Vertebral Body Height during Follow-up? A Multiple Linear Regression Study"

_geriatrics, 2022, doi:10.3390/geriatrics7060142_

Round 1
Reviewer 1 Report
The aims of this study are to evaluate 1) the vertebral body height before and following percutaneous vertebroplasty for osteoporotic vertebral compression fracture; 2) factors influencing restoration of vertebral body height at different time points (pre-surgery, pre-cementation, post-cementation, 6-week follow-up, and the last follow (minimum 6 months)). Forty-eight patients (108 vertebrae) were included. The vertebral body height changed from 2.2 cm at baseline (standing) to 2.3 cm at pre-cementation (prone), 2.5 cm at post-cementation, 2.3 cm at 6-week follow-up (standing), and 2.2 cm at the last follow-up.
Page 2, Lines 71-74
The authors wrote 28 vertebrae were treated with percutaneous vertebroplasty and 20 with a flexible trocar vertebral augmentation system. I believe the authors meant to say 28 and 20 "patients" (instead of vertebrae).
Page 9, Line 220
The author wrote this is a prospective study, but nothing was mentioned in the method session. Is this a retrospective review or a prospective study? There are two treatment groups (percutaneous vertebroplasty versus StabiliT system). Are the patients randomized to one of the treatment modalities?
Author Response
Reviewer #1:
The aims of this study are to evaluate 1) the vertebral body height before and following percutaneous vertebroplasty for osteoporotic vertebral compression fracture; 2) factors influencing restoration of vertebral body height at different time points (pre-surgery, pre-cementation, post-cementation, 6-week follow-up, and the last follow (minimum 6 months)). Forty-eight patients (108 vertebrae) were included. The vertebral body height changed from 2.2 cm at baseline (standing) to 2.3 cm at pre-cementation (prone), 2.5 cm at post-cementation, 2.3 cm at 6-week follow-up (standing), and 2.2 cm at the last follow-up.
My comments are:
Point 1. Page 2, Lines 71-74. The authors wrote 28 vertebrae were treated with percutaneous vertebroplasty and 20 with a flexible trocar vertebral augmentation system. I believe the authors meant to say 28 and 20 "patients" (instead of vertebrae).
We have changed this paragraph and we have specified the number of patients, technique used and vertebrae treated. We added: “Finally, 48 patients were included. Thirty-four patients (78 vertebrae) were treated with percutaneous vertebroplasty (DePuy Synthes, Oberdorf, Switzerland), ten patients (16 vertebrae) were treated with a flexible trocar StabiliT® MX Vertebral Augmentation System (DFine, San José, USA) and four patients were treated with both techniques (10 vertebrae with percutaneous vertebroplasty and 4 vertebrae with StabiliT® system)”
Point 2. Page 9, Line 220. The author wrote this is a prospective study, but nothing was mentioned in the method session. Is this a retrospective review or a prospective study? There are two treatment groups (percutaneous vertebroplasty versus StabiliT system). Are the patients randomized to one of the treatment modalities?
We included the study type in method section. We added “We conducted this observational, prospective and non-randomized study in compliance with the principles of the Declaration of Helsinki. The study’s protocol was reviewed and approved by the Institutional Review Board of University of Navarra (IRB No. 2018.044). Written informed consents were obtained in all patients.”.

Reviewer 2 Report
Dear authors,
thank you very much for giving me the opportunity to review your paper. It adresses an important topic in the everyday treatment of osteoporotic vertebral fractures. I would recommend to add a few sentences of the impact of the non-restored vertebral body height on the sagittal balance; numerous papers have adressed this topic and I somehow miss it here.
Yours sincerely
Author Response
Response to reviewer 2.
Geriatrics-2055635: Vertebral body height changes in acute symptomatic osteoporotic vertebral compression fractures treated with vertebral cement augmentation. Which factors affect vertebral body height during follow-up? A multiple linear regression study.
We have written our responses to the reviewer comments, point by point, in blue.
Reviewer #2:
Thank you very much for giving me the opportunity to review your paper. It adresses an important topic in the everyday treatment of osteoporotic vertebral fractures. I would recommend to add a few sentences of the impact of the non-restored vertebral body height on the sagittal balance; numerous papers have adressed this topic and I somehow miss it here.
We added one paragraph at the end of introduction section about sagittal balance and osteoporotic vertebral fractures. “For years, the relationship between osteoporotic vertebral fractures and sagittal balance changes has been proven. Isolated vertebral compression fractures change local parameters. However, if there are multiple fractures, they might cause sagittal spinal imbalance [17,18]. Dorsal hyperkyphosis and loss of vertebral height create an anterior displacement of the gravity center. These alterations might cause respiratory problems, gastrointestinal disturbances and, ultimately, spinal canal compression. In the anterior body vertebral segment, a flexion moment significant is created, the posterior muscles and ligaments must compensate for this flexion excess and the anterior body vertebral segment weakened must resist more compressive forces. This leads to impaired balance, increased muscle fatigue, reduced gait speed, increased risk of falls and, therefore, increased risk of additional fractures [19]. A recent study by Su et al. [20] studied the effect of PVP on sagittal alignment in 42 patients. They found that two weeks after treatment pain, function and sagittal balance improved: sagittal vertical axis (49 to 37.75), T1 pelvic angle (22,98 to 20.69) and thoracic kyphosis (25.83 to 14.91). It is also important to be aware that the previous presence of sagittal imbalance is a risk factor to vertebral re-collapse after PVP [21].
